# Anti-Tumor and Anti-Inflammatory Activity In Vivo of *Apodanthera congestiflora* Cogn. (Cucurbitaceae)

**DOI:** 10.3390/pharmaceutics13050743

**Published:** 2021-05-18

**Authors:** Geovana F. G. Silvestre, Renally P. Lucena, Genil D. Oliveira, Helimarcos N. Pereira, Jhonatta A. B. Dias, Ivone A. Souza, Harley S. Alves

**Affiliations:** 1Post-graduation Program of Pharmaceuticals Sciences, State University of Paraíba, Campina Grande 58429-500, Brazil; geovana.silvestre@aluno.uepb.edu.br; 2Department of Pharmacy, State University of Paraíba, Campina Grande 58429-500, Brazil; renally.lucena@aluno.uepb.edu.br (R.P.L.); genil.oliveira@aluno.uepb.edu.br (G.D.O.); 3Department of Antibiotics, Federal University of Pernambuco, Recife 50670-901, Brazil; helimarcosnunes@servidor.uepb.edu.br (H.N.P.); jhonatta.dias@ufpe.br (J.A.B.D.); ivone.souza@ufpe.br (I.A.S.)

**Keywords:** Cucurbitaceae, norcucurbitacins, acute toxicity, anti-inflammatory, Ehrlich’s carcinoma

## Abstract

This work aimed to carry out a study of *Apodanthera congestiflora* by investigating its chemical composition and pharmacological potential. From the dichloromethane phase (Dic-Ac) of the *A. congestiflora* stems, three compounds were identified: cayaponoside C_5b_ (Ac-1), cabenoside C (Ac-2) and fevicordin C_2_ glucoside (Ac-3), being last identified for the first time as a natural product. These compounds were obtained by chromatographic methods and their structures were elucidated by means of spectroscopic analysis of IR, MS and NMR. In the quantification of Dic-Ac, it was possible to observe the presence of 7% of cayaponoside C_5b_. Dic-Ac showed significant toxicity for in vivo tests, with macroscopic and biochemical changes. The anti-inflammatory activity of Dic-Ac was investigated using the paw edema model. A decrease in inflammatory signs was observed in the first 5 h and the most effective dose in reducing edema with was 7.5 mg kg^−1^ (66.6%). Anti-tumor activity of Dic-Ac was evaluated by Ehrlich’s carcinoma model, which showed inhibition rate of 78.46% at 15 mg kg^−1^ dosage. The phytochemical investigation, together with the biological tests carried out in this study, demonstrated that *A. congestiflora* is a promising species in the search for therapeutics, since it contains substances with high pharmacological potential in its composition.

## 1. Introduction

With worldwide distribution and a high number of species, the family Cucurbitaceae Juss. is considered one of the most important families in the world of flora [1]. In Brazil, 30 genera and 160 species are registered and distributed in all regions of the country [2]. According to Silva et al. (2016) [3], cucurbits have considerable economic importance, due to the consumption of their vegetables that are produced and valued by the population, as is the case of the variant species of melon (*Cucumis melo* L.), watermelon (*Citrullus lanatus* (Thunb Matsum & Nakai)) and pumpkins (*Cucurbita* sp.).

In addition to the nutritional use of its fruits, ethnobotanical records point to the use of several species of the Cucurbitaceae family in popular medicine, as well as showing the presence of compounds with a broad biological spectrum, such as cucurbitacins. This group of secondary metabolites has great chemical and biological relevance, as it presents a vast structural diversity and pharmacological potential. The anti-inflammatory, antidiabetic, antiulcerogenic, analgesic, antiparasitic and antitumor properties can be cited [4,5].

Although cucurbitacins have a wide biopharmaceutical spectrum, most investigations involve anti-tumor activity [6]. The pioneering studies that relate cucurbitacins to an anti-tumor potential date from the 1960s, which made the class known for presenting this property. This fact can support the use of plants that contain them for various purposes that involve, directly or indirectly, cell death [4,5]. According to Ali et al. (2019) [7], cucurbitacins was able to treat cancers such as human lung adenocarcinoma, glioblastoma multiform, nasopharyngeal carcinoma, or chronic lymphocytic leukemia. The anti-inflammatory activity was mentioned by Ríos, Escandell and Recio (2005) [8] and Kaushik et al. (2015) [9], who pointed out that these substances can act in different signaling pathways of the immune system.

Madaleno (2011) [10] and Roque and Loiola (2013) [11] observed that in the Caatinga of Northeast Brazil, the species *Apodanthera congestiflora* is one of the Cucurbits most cited in studies of species for medicinal purposes. Named by Martínez Crovetto in 1954 as *Melothria congestiflora* Mart. Crov., *A. congestiflora* is a woody vine endemic to Brazil. Popularly known as “teiú” and “cabeça-de-nego”, it is used in popular medicine mainly in the form of tea/licking to relieve pain in the spine and as a blood purifier (“thick blood”). The tea prepared with the roots is also indicated for external use to treat skin blemishes and itchiness [12].

Even though the species *A. congestiflora* is mentioned in the literature as a medicinal plant, the absence of studies related to the species is a factor that motivates chemical–pharmacological research focusing on the discovery of therapeutic agents and the development of new drugs. So far, reports about this species are scarce and refer only to the roots [13,14]. The objective of this research was to conduct a chemical–pharmacological study of the *A. congestiflora* stems, promoting the identification of chemical compounds as well as assessing the safety of use, anti-tumor and anti-inflammatory activity of extracts derived from the species.

## 2. Materials and Methods

### 2.1. Plant Material

The stems from *A. congestiflora* were collected from adult plants in the municipality of Barra de Santana, in the Paraíba semi-arid region (6°43′18″ S–36°03′46″ W), which were subjected to selection, cutting and cleaning processes, with branches containing reproductive parts of the plant being reserved for the making and storage of exsiccate in the herbarium Manuel de Arruda Câmara—number 1000 of the State University of Paraíba, Campina Grande—PB. The drying was carried out in an oven with renewal and air circulation, at a temperature of 40 °C, until moisture stabilization was reached. After drying, the material was pulverized in a knife mill, with a defined granulometry of around 10 mesh. Then, the plant drug was packaged in a hermetically sealed bottle, protected from air and solar radiation.

### 2.2. Extaction and Liquid: Liquid Partition

The plant material (1.5 kg) was subjected to extraction by maceration with an ethanol solution at 70%. Twelve extractions were performed using 5 L of solvent at each change, within an interval of 72 h. The obtained macerate went through a process of alcohol evaporation, under reduced pressure, in a rotary evaporator TE-211 (Tecnal^®^, Piracicaba, SP, Brazil) at a temperature of 50 °C. Then, the extractive solution was subjected to drying in a lyophilizer model LS3000 (Terroni, São Carlos, SP, Brazil) providing 233 g (15.53%) of hydroethanolic extract (HE-Ac).

Part of the HE-Ac (200 g) was solubilized in a methanol:water solution (7:3—*v*:*v*) and partitioned in organic solvents providing 12.89 g of the hexane (Hex-Ac), 31.18 g of the dichloromethane (Dic-Ac), 19.41 g of ethyl acetate (AcOEt-Ac) and 3.8 g of *n*-butanol (*n*-BuOH) phases.

### 2.3. Isolation of Chemical Compounds

Dic-Ac (8.5 g) was chromatographed on a column (C1) with silica gel 60 (Macherey-Nagel, Düren, Germany) providing 58 fractions. Fraction 35 (500 mg) of C1 showed a yellowish precipitate soluble in methanol and was purified in a column with Sephadex LH-20 (SIGMA-ALDRICH^®^, Saint Louis, MO, USA) and MeOH:CHCl_3_ solution (1:1—*v*:*v*), yielding a mixture with 3 compounds (Ac-1), (Ac-2) and (Ac-3). The chromatographic process of the Dic-Ac is shown in Scheme 1.

### 2.4. Spectral Analysis

Fourier transform infrared spectroscopy (FTIR) was performed on FT-IR Spectrometer 400 equipment (PerkinElmer^®^, Waltham, MA, USA), using KBr inserts for dispersing the samples and a scanning range between 4000 to 650 cm^−1^. The nuclear magnetic resonance spectra of hydrogen (^1^H NMR) and carbon (^13^C NMR), were recorded in a BRUKER AVANCE III spectrometer (Bruker, Billerica, MA, USA) operating at 400 MHz for ^1^H and 100 MHz for ¹³C, using the attached proton test (APT), heteronuclear single quantum coherence spectroscopy (HSQC) and heteronuclear multiple bond correlation (HMBC). The deuterated solvents used were CDCl_3_ and acetone-d_6_. The chemical shifts (δ) were expressed in parts per million (ppm), using the solvent itself as an internal reference and the coupling constants (J) were given in Hz. The mass spectra were obtained on ESI-ITMS, model Amazon X (Bruker, Billerica, MA, USA).

#### 2.4.1. Cayaponoside C_5b_ (Ac-1)

Yellow amorphous powder (86 mg—in mixture); IR (KBr) ν_max_: 3042, 2970, 2924, 1685, 1485, 1388, 1300, 1076, 1026 cm^−1^. ^1^H and ^13^C NMR (Acetone-d_6_, 400 and 100 MHz) spectroscopic data, see Table 1; ESIMS *m*/*z* 661.26 ([M–H]^−^, (C_35_H_50_O_12_).

#### 2.4.2. Cabenoside C (Ac-2)

Yellow amorphous powder (86 mg—in mixture); ^1^H and ^13^C NMR (Acetone-d_6_, 400 and 100 MHz) spectroscopic data, see Table 1; ESIMS *m*/*z* 545.06 ([M–H]^−^, (C_29_H_38_O_10_).

#### 2.4.3. Fevicordin C_2_ Glucoside (Ac-3)

Yellow amorphous powder (86 mg—in mixture); ^1^H and ^13^C NMR (Acetone-d_6_, 400 and 100 MHz) spectroscopic data, see Table 1; ESIMS *m*/*z* 659.15 ([M–H]^−^, (C_35_H_48_O_12_).

### 2.5. Quantification of Cayaponoside C_5b_ by Nuclear Magnetic Resonance (^1^H-NMR)

The acquisition of the ^1^H-NMR spectrum was performed on a Bruker Avance Neo 500 instrument operating at 500 MHz (Bruker, Billerica, MA, USA). To obtain the ^1^H NMR spectra, the following parameters were used: solvent acetone-d_6_; temperature: 25 °C; number of scans: 64; receiver gain: auto; acquisition time: 2 min. For quantification, TopSpin Eretic 2 software was used, which was calibrated using ethyl benzene (Bruker, Billerica, MA, USA) (8.16 mM) used as an external standard, using the same parameters described above. In the quantification, the peak area was used, which had the initial and final points of integration done manually. The concentration of Dic-Ac phase used to perform the tests was 20.45 mg mL^−1^. The experiment was carried out in triplicate. The value of cayaponoside C_5b_ present in the Dic-Ac is expressed in %weight of cayaponoside C_5b_ ± standard deviation/weight of Dic-Ac phase (*% w*/*w*).

### 2.6. Toxicity in Human Erythrocytes

The hemolytic activity assay was based on the method by Pinto et al. (2012) [15] with adaptations. To carry out this experiment, freshly collected blood samples of type O^+^ were collected in tubes containing EDTA anticoagulant. The erythrocytes were separated by centrifugation (3000 rpm, 5 min), washed with saline and centrifuged again. This process was repeated 3 times for the complete removal of the plasma. After washing, the erythrocytes were resuspended in saline solution at a concentration of 1%. The Hex-Ac and Dic-Ac samples were dissolved in saline and 0.2% Tween 80™ at concentrations of 10, 100 and 1000 µg mL^−1^. A mixture of 2 mL of the red blood cell suspension + 500 µL of the test solutions was made and left for 1 h at room temperature for hemolysis to occur. Concomitantly, negative control (saline + 0.2% Tween 80™) and positive control (1% Triton X-100™) were used for hemolysis.

After the incubation period, the samples were centrifuged and the overflowing liquid was subjected to analysis in a UV-VIS SHIMADZU spectrophotometer (Shimadzu, Kyoto, Japan) at 540 nm wavelength. The hemolytic potential of the tested samples was calculated from the following Equation (1):Hp *(%)* = [(Ea − Ba)/Ta] × 100%,(1)
where Hp = Hemolytic potential (%); Ea = Extract absorbance; Ba = Blank absorbance (negative control); Ta = Triton-X 100 absorbance (positive control).

### 2.7. Biological Assays

#### 2.7.1. Experimental Animals and Ethical Procedures

In this study, adult (about 60 days old) albino Swiss mice (*Mus musculus*) were used, weighing between 25 and 35 g—from the State University of Paraíba’s bioterium—and acclimatized for 15 days in the Laboratory of Pharmacological Tests, where they were used to carry out the acute toxicity and inflammation experiments.

The animals were housed in polyethylene cages with stainless steel grids and wood shavings as a cover, with free access to water and balanced feed, kept in an environment with a temperature of 22 ± 2 °C and controlled light, providing a light–dark 12 h cycle. All animals were fasted, with the food removed about 4 h before the beginning of the experiments. During the tests, the animals had free access to water intake. The animals were kept in accordance with the international standards of the Experimental Animal Laboratory Council (ICLAS).

All experiments were carried out in accordance with the standards established by the Brazilian Society for Laboratory Animal Science (SBCAL) and with the standards established by the National Institute of Health Guide for Care and Use of Laboratory Animals. The Ethics Committee on Use in Animals (CEUA-UEPB) approved this study, which was registered as process No. 002/2020 approved on 14 October 2020.

#### 2.7.2. Acute Toxicity

The assessment of acute toxicity was employed using the methodology recommended by the Organization for Economic Cooperation and Development—Guideline 423 [16]. Female mice (60 days) were randomly assigned to two groups of three animals. A single dose was administered orally to the animals in the test group. The control group (*n* = 3) received the vehicle (water + 5% Tween 80™—*v*:*v*) and the treated group (*n* = 3) received Dic-Ac at a dose of 2000 mg kg^−1^.

The animals were observed in the first two hours and then every 24 h daily, for 14 days, after the administration of Dic-Ac. The evaluation was carried out using the hypocratic screening method and, in addition, the mass, water and feed consumption were evaluated daily. On the 14th day, the animals were anesthetized with ketamine and xylazine (2:1—*v*:*v*) intraperitoneally, and then blood was collected by cardiac puncture for hematological and biochemical tests. Euthanasia of the animals was accomplished using the same combination of anesthetics in a lethal dose. The liver, kidney, spleen, lung and heart organs were collected for macroscopic analysis, as well as for the determination of the organ index calculated according to the formula (2):Relative weight = (Organ weight/Animal weight) × 100%(2)

Subsequently, the entire experiment was repeated according to the same initial conditions to confirm the results obtained. The results found were analyzed and evaluated in the same way and expressed as an average between groups.

#### 2.7.3. Anti-Inflammatory Activity—Carrageenan-Induced Paw Edema

In this test, the animals were divided into 5 groups (*n* = 8). The groups received the substances orally: Dic-Ac at doses of 30, 15 and 7.5 mg kg^−1^, a control group (water + 5% Tween 80™) and a standard drug group (dexamethasone 10 mg kg^−1^).

One hour after the treatment of each group, the inflammatory process was induced through an intraplantar injection (i.pl.) of 50 µL of carrageenan (1%) in the right hind paw. The animal’s other paw was also analyzed for volume measurement and compared with the paws that received carrageenan. Paw volumes were measured before induction by the phlogistic agent and, after induction, at pre-established intervals of 1, 2, 3, 4 and 5 h after the administration of carrageenan. The volume of the edema (mL) was recorded using a plethysmometer (Ugo Basile^®^, Gemonio, Italy). The animal’s posterior paw was submerged until the tibio-tarsal junction in the device’s reading chamber. The volume of liquid displaced was recorded digitally and corresponded to the volume of the paw. The results were expressed as the difference in volume (mL) between the paw that received carrageenan and the contralateral paw that did not receive carrageenan.

#### 2.7.4. Anti-Tumor Activity in Ehrlich’s Carcinoma

In this experiment, tumor cells from the Ehrlich Carcinoma lineage (solid form) were removed from donor animals with eight days of implantation, through ascites aspiration and introduced into the recipient animals, subcutaneously in the right axillary region, in a concentration of 2.5 × 10^7^ cells/mL [17]. The recipient animals were divided into four groups (*n* = 5) subdivided into a control group (0.9% saline), a standard group (cisplatin 2.5 mg kg^−1^) and treated groups (Dic-Ac 15 mg kg^−1^ and 30 mg kg^−1^). Treatment was started after 24 h of the tumor’s implantation and the compounds’ tests were administered orally for seven days. On the eighth day, the animals were sacrificed in a CO_2_ chamber and the tumors were removed and separated by the group, dissected, weighed and then the difference between the control, standard and treated groups was calculated to obtain the tumor inhibition index according to the Equation (3):TWI% = [C − T/C] × 100%(3)
where TWI % = percentage of tumor growth inhibition, C = average tumor weight of animals in the control group, T = average weight of animals in the treated group [18].

#### 2.7.5. Data Analysis

The toxicity experiments were statistically evaluated through *t* test, with a 95% confidence interval, using the Graph Pad Prism 8.0 software. Values of “*p*” less than 0.05 (*p* < 0.05) were considered as indicative of significance. The carrageenan-induced paw edema experiments were evaluated using two-way analysis of variance (ANOVA), followed by the Bonferroni test with a 95% confidence interval, using the Graph Pad Prism 5.0 software. Values of “*p*” less than 0.05 (*p* < 0.05) were considered as indicative of significance. The Ehrlich’s carcinoma experiments were evaluated using the statistical analysis using two-way ANOVA followed by the Tukey’s multiple comparison test.

## 3. Results and Discussion

### 3.1. Caracterization of Mixture Ac-1, Ac-2 e Ac-3

The IR spectrum (Appendix A) showed a wide band at 3402 cm^−1^ characteristic of O–H stretch, bands at 2970 and 2924 cm^−1^ typical of C–H bonding of carbon sp^3^, in addition to a medium intensity band in 1685 cm^−1^ suggesting the presence of ketone carbonyl. The two bands at 1485 and 1388 cm^−1^ indicate bend of the C–H bond, besides a band at 1076 cm^−1^ suggestive of the C–O bond [19].

In the ^1^H NMR data (*δ*, Acetone-d6, 400 MHz) a signal envelope between *δ*_H_ 2.5–0.95 (Appendix A) suggestive of hydrogens linked to sp^3^ carbon was visualized, of which seven of them were associated with groups methyl at *δ*_H_ 0.95 (C-18); 1.20 (C-19); 1.38 (C-21); 1.13 (C-26); 1.12 (C-27); 2.21 (C-28) and 1.01 (C-30). In the region between *δ*_H_ 5.80–6.94 (Appendix A), typical signs of protons linked to sp^2^ carbons were seen. In addition, singlets in *δ*_H_ 6.60, 6.58 and 6.55 suggest the presence of aromatic hydrogens present in norcucurbitacin structures [20]. Through the observation of the integral of these last signs, it was possible to define that the sample was a mixture of three compounds.

The expansion of ^1^H NMR spectrum (Appendix A) showed the presence of the signals between *δ*_H_ 3.59 and 3.40 (m, H-2’, H-3’, H-4’ and H-5’) indicative of an osidic unit, in addition to a doublet at *δ*_H_ 4.65 (*J* = 7.4 Hz, H-1‘) characteristic of anomeric proton. Such data suggest the presence of a *β*-glucose type sugar molecule [20,21].

The ^13^C NMR spectrum (Appendix A) allowed the visualization of approximately 80 signals, which confirmed the presence of more than one compound in the analyzed fraction. Two signals were observed in the low field (*δ*_C_ 213.8 and 212.7—Appendix A) suggestive of ketone carbonyls present in tetracyclic triterpenes of the cucurbitan nucleus [22] corroborating the IR spectrum. It was also possible to note signs in the aromatic ring region at *δ*_C_ 112.8; 145.2; 144.9; 122.3; 129.3 and 129.4 (Appendix A) referring to the carbons of ring A of the tetracyclic nucleus C-1, C-2, C-3, C-4, C-5 and C-10, in addition to signals between *δ*_C_ 70.3–77.7 (Appendix A) that are typical of sugar carbons, which reinforces the presence of a glycosidic unit linked to a triterpene nucleus.

A signal was observed at approximately *δ*_C_ 58.7 (Appendix A) which is characteristic of C-17 with a side chain directly connected to it [20]. In this side chain, the signals at *δ*_C_ 29.6 (C-26) and *δ*_C_ 29.4 (C-27) (Appendix A), suggestive of methyl carbons, and a signal at *δ*_C_ 79.6 (C-25) was identified, making it possible to deduce that C-25 supported two methyl and one hydroxyl [23]. These signals were attributed to the major compound called the cayaponoside C_5b_ (Ac-1). However, a signal at *δ*_C_ 67.5 (Appendix A) referring to carbon 17 was also seen, when it does not support a side chain and another signal at *δ*_C_ 208.6 compatible with ketone carbon at position C-20 [24]. The presence of this value for carbon 17 was reinforced by the interactions seen in the HSQC heteronuclear correlation map (Appendix A) between *δ*_H_ 3.07/*δ*_C_ 67.8 and in HMBC (Appendix A) between *δ*_H_ 3.07/*δ*_C_ 49.5, 72.0, 208.4 and 19.7 confirming that the second compound did not have the side chain and that it was the cabenoside C (Ac-2). The molecular formulas C_35_H_50_O_12_ and C_29_H_38_O_10_ were determined by negative-mode ESI-MS measurement of the molecular ions at m/z 661.26 [M–H]^−^ and 545.06 [M–H]^−^ of Ac-1 and Ac-2, respectively (Appendix A).

For the carbons C-23 and C-24 of the side chain of the cayaponoside C_5b_ (Ac-2) the correlations between *δ*_H_ 2.85/*δ*_C_ 32.1 and *δ*_H_ 1.69/*δ*_C_ 38.0, in the HSQC (Appendix A) and between *δ*_H_ 2.86/*δ*_C_ 37.7; 215.3 and *δ*_H_ 1.69/*δ*_C_ 215.4; 32.0 and 29.3, in HMBC (Appendix A), respectively, were visualized, confirming that these two carbons have sp^3^ hybridization and that the values *δ*_C_ in *δ*_C_ 32.0, 37.7, 215.4, and 29.3 can be attributed to the carbons of positions 23, 24, 22 and 26–27, respectively. However, the occurrence of the signals at *δ*_C_ 120.6 and 155.3 points to the presence of unsaturated carbons at positions 23 and 24 indicating the presence of an isomer of the compound Ac-2, not yet evidenced in the literature.

Two doublets in the ^1^H NMR spectrum were recorded at *δ*_H_ 6.81 and 6.94, with coupling constants of *J* = 15.4 Hz, together with the signal at *δ*_C_ 203.6 seen in ^13^C NMR, suggesting the presence of vinyl hydrogens in an α, β-unsaturated carbonyl. To support these data, the signals between *δ*_H_ 6.81/*δ*_C_ 120.6 and *δ*_H_ 6.94/*δ*_C_ 155.3 were also visualized in the heteronuclear correlation map—HSQC, and in the HMBC, the couplings between *δ*_H_ 6.81/ *δ*_C_ 70.4, 155.2 and 203.5 and *δ*_H_ 6.94/*δ*_C_ 70.4, 120.3, 203.3 and 29.4, confirming 30.4, 70.4, 120.3, 155.2 and 203.3 for positions 26, 25, 23, 24 and 22, respectively. The molecular formula C_35_H_48_O_12_ was determined by ESIMS at m/z 659.15 [M–H]^−^ and confirmed Ac-3 (Appendix A).

According to the interpretation of the IR, MS and NMR signals, using uni and bidimensional techniques and, aided by the comparison with the data described in the literature [20,21,23], it was possible to deduce that the mixture of the compounds contained norcucurbitacin, which have an aromatic ring, commonly designated as a 2,3-di-*O*-substituted derivative. Three compounds were identified in this mixture (Figure 1), two of which are already known in the literature: cayaponoside C_5b_ and cabenoside C, as well as a new 29-norcucurbitacin named of fevicordin C2 glycoside. Norcucurbitacins are a subclass of cucurbitacins and have structures known for the absence of methyl at position C-28 or C-29. They have already been shown in species of the genus *Fevillea, Cayaponia* and *Wilbrandia*, all belonging to the family Cucurbitaceae [25], but identified for the first time in the genus *Apodanthera*.

The Cucurbitaceae family is known for the variety of secondary metabolites it produces, especially cucurbitacins, as demonstrated by Kaushik et al. (2015) [9]. Previous studies carried out with aqueous extracts from the vegetable drug of *A. congestiflora*, in different particle sizes, showed a high concentration of saponins (18.9–37.5%) in the spectroscopic analysis made in a spectrophotometer using diosgenin as a standard [26]. However, no study was carried out demonstrating the concentration of isolated compounds present in the extracts of species of the genus *Apodanthera*. Thus, we performed the absolute quantification of the cayaponoside C_5b_, present in the dichloromethane phase, using the TopicSpin Eretic 2 method from the studies of evaluation of ^1^H NMR spectra [27].

Initially, the signals seen in the ^1^H NMR spectrum of the dichloromethane phase that referred to the cayaponoside C_5b_ were assigned (Appendix A). The typical signal at *δ*_H_ 6.60 of the aromatic A ring, the signals at *δ*_H_ 6.87 (d, 10.0 Hz, 1H) and *δ*_H_ 5.50 (dd, 6.2 and 10.0 Hz, 1H) referring to the olefinic protons in C-6 and C-7, respectively, in addition to the terminal methyls at *δ*_H_ 1.12 (s, 3H) and *δ*_H_ 1.13 (s, 3H) were visualized for Ac-1. The signal at *δ*_H_ 1.13 (Appendix A) was used to quantify the cayaponoside C_5b_ in the dichloromethane extract, as this signal did not show any overlap with other signals in the ^1^H NMR spectrum. Thus, in this experiment we were able to reveal that this compound represents about 7% (SD = 0.02 and CV = 1.45) of the dichloromethane phase of *A. congestiflora (% w*/*w*).

### 3.2. Hemolytic Activity

The potential of Dic-Ac to promote damage to the erythrocyte membrane was assessed by the hemolytic assay, which allows investigating the ability of a product to promote the formation of pores or cause rupture of the erythrocyte membrane. Hemolysis did not occur for concentrations of 10 and 100 µg mL^−1^. Therefore, the hemolytic potential (Hp) was calculated only for the concentration of 1000 µg mL^−1^, with Hp = 0.28% for Dic-Ac.

Although these studies report low or no toxicity, the fact that saponins are present in the extracts could lead to greater toxicity, since saponins, due to their amphiphilic character, generally present interactions with cell membranes, including hemolytic and ichthyotoxic action [28].

The hemolytic potential of the roots from *A. Congestiflora* was assessed by Pereira (2017) [13] and Videres (2017) [14], using the similar assay. In both, no evidence of hemolysis was found in the tested extracts.

### 3.3. Acute Toxicity

The animals were treated with Dic-Ac at a dose of 2000 mg kg^−1^ orally, in view of the finding of signs of toxicity by the animals in a study by Pereira (2017) [29], whereby the ethyl acetate phase was obtained from the roots of the plant.

In the first stage of the evaluation, called the hypocratic test, clinical signs of toxicity were observed. Therefore, the animals showed several behavioral changes, especially 4 h after the administration of the phase, such as: increased urination and defecation, loss of straightening reflex, ataxia, eyelid ptosis, tachycardia and respiratory changes. In addition, the Dic-Ac caused mortality of an animal within 24 h. This last event motivated the repetition of the experiment, according to the OECD protocol 423 [16], with a death being confirmed again over the 14 days.

In Table 2, an increase in water consumption and a reduction in feed consumption by the animals tested can be observed in relation to those in the control group, suggesting signs of toxicity for consideration, albeit discrete, since the animals did not present weight gain in the same proportion as those of the control group.

The reduction in feed consumption by the group treated with Dic-Ac corroborates the data on the final weight of the animals and denotes a lower weight gain than that obtained by the control group over the 14 days of the experiment, suggesting a possible toxicity linked to the product in the tested dose. This fact can be explained by the ability of some toxic active ingredients to cause apathy, anorexia and diarrhea, symptoms that can lead to weight loss or progressive weight loss in exposed individuals [30].

In this regard, throughout the chemical investigation of Dic-Ac carried out in the present work, 3 cucurbitacins were isolated (Ac-1, Ac-2 and Ac-3). Most compounds belonging to this class have high toxicity and a wide range of biological activities, including cytotoxic activity [24]. The presence of these constituents may explain the relative toxicity of Dic-Ac, although in vitro tests for hemolytic activity in this phase have not shown significant toxicity.

There was also a significant increase in the size and mass of the liver of the animals treated with Dic-Ac in the dose used (Table 3), suggestive of hepatomegaly caused by the constituents present. We can suggest that in the dose used, Dic-Ac may be involved in metabolic pathways that are directly linked to the liver, since it is responsible for the metabolism of various endogenous and exogenous substances [31]. Furthermore, an increase in serum levels of transaminases was also observed (Table 4), important enzymes in liver damage events, corroborating the hepatomegaly found in the macroscopic and quantitative evaluation.

The toxicity shown in the in vivo assay may not be directly related to red cell toxicity. This proposition was confirmed by the complete blood count of the animals’ peripheral blood (Table 4), which showed no changes in hematological parameters, corroborating the data obtained in vitro in the hemolysis test.

Different to the results of the macroscopic analysis of the liver of animals submitted to the acute toxicity test, no significant changes were found in the evaluation of other liver enzymes and metabolites directly involved in hepatic pathways, such as Gamma-GT and bilirubin. Significant changes in glucose and total cholesterol levels were only seen in the treated group compared to the control. In this sense, it is necessary to carry out more specific assessments of liver tissue lesions, such as immunohistochemical and ultrastructural assessment of the lesions, as they are techniques that exhibit a wide dimension of the degree of hepatic impairment induced by natural products [32].

The toxicity of an exogenous compound can manifest itself through cell damage in different tissue structures, triggering physiological changes, mainly in liver and kidney function. Metabolism in general is also susceptible to changes, which were observed in this study through hematological and biochemical parameters. It is for this reason that the analysis of the toxicity of any substance is a complex and multifactorial process, since the manifestation of the damage can be silent, apparent or even expressed due to multiple exposures to the supposedly toxic agent. Because of these factors, this experiment is an initial and preliminary step in the study of the toxicity of any agent, including those of natural origin [16].

Therefore, based on these premises, it is not possible to establish or seal the toxicity of a substance in a simple way. In this sense, despite the apparent preliminary liver damage, there were no significant changes in the other organs analyzed and in most serological tests, mainly in renal function.

### 3.4. Anti-Inflammatory Activity

To evaluate the anti-inflammatory potential of dichloromethane phase (Dic-Ac), the carrageenan-induced paw edema model at doses of 7.5, 15 and 30 mg kg^−1^ per animal mass was used. The doses chosen were based on experiments previously carried out by our research group, considering that from the assessment of acute toxicity at the dose of 2000 mg kg^−1^, the death of an animal was found. The experiment was repeated with a dose of 300 mg kg^−1^, which corroborated the result obtained in the previous test. Thus, it was chosen as the highest dose to be tested at 30 mg kg^−1^, corresponding to 10% of the second highest dose recommended by the OECD protocol.

It was observed that the Dic-Ac reduced edema in a continuous and sustained way during the 5 h of observation, with no statistical difference between the doses tested (Figure 2). In this perspective, it is suggested that the effect of this phase is not dose-dependent, something positive, as this fact allows a pronounced anti-inflammatory effect to be obtained even with the administration of low doses, which would supposedly lead to a lower risk of toxicity.

The highest percentage of edema inhibition (Table 5) occurred after 2 h, at the dose of 7.5 mg kg^−1^ (66.6%), a fact that was repeated in a similar way until the fourth hour of observation. In the fifth hour, the most effective dose was 30 mg kg^−1^ (64.3%). This phenomenon can be explained by the fact that the administration of the highest dose is able to maintain a sufficient plasma concentration to maintain the inhibition of inflammation for a longer time. In this case, a further evaluation of the pharmacokinetics of that phase is necessary to verify and validate these preliminary observations.

The anti-inflammatory activity of *Apodanthera congestiflora* has been cited by ethnopharmacological surveys and in previous studies carried out with its roots, which also validated popular knowledge about this species [15,16]. It is worth noting that the three norcucurbitacins evidenced in the study of chemical bioprospecting were present in the Dic-Ac and may be involved in the anti-inflammatory activity observed in pharmacological tests [20,23].

Peters et al. (2003) [33] observed that the dichloromethane extract of *Wilbrandia ebracteate* (Curubitaceae) significantly inhibited inflammatory parameters in experimental models of in vivo and in vitro inflammation. An analysis of the chemical composition of the extract via HPLC indicated the presence of cucurbitacins in the extract.

### 3.5. Anti-Tumor Activity

The results were evaluated using the Dic-Ac. After checking for seven days and compared to the traditional treatment with Cisplatin antineoplastic agent, results showed a significantly reduction in tumor weights (Figure 3). The oral solutions of Dic-Ac presented anti-tumor activity by inhibition growth of TW% 76.05% and 78.46% (Table 6), with total remission of the tumor on the seventh day of administration, which leads us to consider its importance ahead of usual therapies, which have side effects and are more costly.

During the tests, there was no death of any animals or behavioral changes such as arching gait, loss of muscle mass, stable dynamics, hair loss and loss of appetite.

The presence of cucurbitacins in the Dic-Ac suggests that these substances can play a fundamental role in reducing tumors. According to Liang and Dan (2019) [6] these compounds can exert antiproliferative effects against various cancers, acting by mechanisms such as inducing apoptosis, cell cycle arrest and autophagy. An in vivo study carried out with cucurbitacins E and I glucosides [34], also performed with Ehrlich’s ascites carcinoma model, indicated that these compounds were able to decrease tumor size, prolong the survival time of the mice as the life expectancy, as well as normalize biochemical parameters of infected animals.

## 4. Conclusions

The results obtained showed that *Apodanthera congestiflora* is a relevant species in the search for pharmacologically active compounds. From the dichloromethane phase, 3 compounds were obtained. The spectroscopic analysis allowed the identification of them as being two triterpene saponins of cucurbitan nucleus belonging to the class of 29-norcucurbitacins and the identification as cayaponoside C_5b_ (Ac-1) and cabenoside C (Ac-2), obtained for the first time in the genus *Apodanthera*, in addition to a new substance in the same class called fevicordin C2 glycoside (Ac-3).

The hemolysis test indicated low toxic potential, however, the acute toxicity test indicated that Dic-Ac has significant toxicity under the conditions evaluated. It was evidenced that the Dic-Ac has anti-inflammatory activity during the 5 h following the injury (paw edema) induced by carrageenan. The response observed was not dose-dependent and the dose of 7.5 mg kg^−1^ reached a 66.6% reduction in inflammation in the second hour of observation, this being a considerably low dose, considering that it is an organic phase and not of isolated substance.

Dic-Ac was able to significantly decrease volume of tumors induced by Ehrlich’s carcinoma cells in mice. At a dose of 15 mg kg^−1^, the percentage of inhibition was 78.46%, a result that indicates a strong potential of anti-tumor activity. These data reveal that *A. congestiflora* is a source of cucurbitacins and, therefore, it is possible to attribute it to a vast range of pharmacological potentialities.

## Data Availability

Data available on request.

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
