# Peer review of "Anti-Tumor and Anti-Inflammatory Activity In Vivo of Apodanthera congestiflora Cogn. (Cucurbitaceae)"

_pharmaceutics, 2021, doi:10.3390/pharmaceutics13050743_

Round 1
Reviewer 1 Report
The manuscript presented by Silvestre and et al. is well written, clear, and easy to read. It adds information to the area of ethno-pharmacology and elucidates the biochemical action of Apodanthera congestiflora. In particular, by hexane (hex)/dichloromethane (dic) extraction the β-sitosterol (Ac-1), the stigmasterol (Ac-2) and the spinasterol (Ac-3), the cayaponoside C5b (Ac- 4) the cabenoside C (Ac-5), and the fevicordin C2 glycoside (Ac-6). Hex-Ac and Dic-Ac differently affect hemolysis therefore Dic-Ac was used for in vivo assay.
Please adds the tumor excision picture near Figure 4
Author Response
Dear reviewer, we appreciate the encouragement.
Reviewer's remarks: The manuscript presented by Silvestre and et al. is well written, clear, and easy to read. It adds information to the area of ethno-pharmacology and elucidates the biochemical action of Apodanthera congestiflora. In particular, by hexane (hex)/dichloromethane (dic) extraction the β-sitosterol (Ac-1), the stigmasterol (Ac-2) and the spinasterol (Ac-3), the cayaponoside C5b (Ac- 4) the cabenoside C (Ac-5), and the fevicordin C2 glycoside (Ac-6). Hex-Ac and Dic-Ac differently affect hemolysis therefore Dic-Ac was used for in vivo assay.
Author´s response: We appreciate the opinion.
Reviewer's remarks:Please adds the tumor excision picture near Figure 4
Author´s response: Unfortunately we were unable to make a photographic record of the excision of the tumor

Reviewer 2 Report
The authors thoroughly investigated purified compounds they derived from Apodanthera congestiflora. They determined in vitro and in vivo toxicity, anti-inflammatory and anti-cancer properties.
Their work provides new and preliminary useful information to be considered for the future development of this compounds for therapeutic use.
Very minor correction Table 4. Glucose- instead of Glycose
The data presented is original and contributes to the validation of ethnobotanical uses of the plant. Isolation of particular compounds and the search for possible therapeutical applications is welcomed in general and specifically regarding this plant, which has not been investigated in detail. Of course, more work has to be performed to prepare any of these compounds for future applications.
Although the work identifies many compounds, using specific solvents, only one compound is further evaluated. The choice of compound is justified but has to be better explained in the text. The hemolysis toxicity test is appropriate but limited to testing cell lysis. Additional in vitro toxicity tests can be done on PBLs or on cell lines, regarding apoptosis. The assay performed are convincing.
The work is publishable and serves as a basis for several other papers investigating compounds derived of Apodanthera congestiflora.
Author Response
Dear reviewer, we appreciate the suggestions.
Response to the reviewer’s comments:
Reviewer's remarks: The authors thoroughly investigated purified compounds they derived from Apodanthera congestiflora. They determined in vitro and in vivo toxicity, anti-inflammatory and anti-cancer properties.
Reviewer's remarks: Their work provides new and preliminary useful information to be considered for the future development of this compounds for therapeutic use.
Author´s response: We appreciate the opinion.
Reviewer's remarks: Very minor correction Table 4. Glucose- instead of Glycose
Author´s response: REVISED
Reviewer's remarks: The data presented is original and contributes to the validation of ethnobotanical uses of the plant. Isolation of particular compounds and the search for possible therapeutical applications is welcomed in general and specifically regarding this plant, which has not been investigated in detail. Of course, more work has to be performed to prepare any of these compounds for future applications.
Author´s response: We are already continuing the research to isolate more compounds, perform pharmacological tests and evaluate their mechanisms of action
Reviewer's remarks: Although the work identifies many compounds, using specific solvents, only one compound is further evaluated. The choice of compound is justified but has to be better explained in the text. The hemolysis toxicity test is appropriate but limited to testing cell lysis. Additional in vitro toxicity tests can be done on PBLs or on cell lines, regarding apoptosis. The assay performed are convincing. –
Author´s response: We justify in the text why the choice of the dichloromethane phase. The phase has a good concentration of norcucurbitacins and this class of compounds is known to have anti-inflammatory and anticancer effects, which justifies the results we obtained in this study.
Author´s response: Unfortunately, we didn't have time to test the compounds on PBL or other cell lines. But, we will follow the recommendations for further work with these molecules.
Reviewer's remarks: The work is publishable and serves as a basis for several other papers investigating compounds derived of Apodanthera congestiflora.
Author´s response: We appreciate the encouragement

Reviewer 3 Report
This work investigated the chemical composition, toxicity, and pharmacological potential, including anti-inflammatory and anti-tumor activity, of Apodanthera congestiflora. The hydroethanolic extract of its stem (HE-Ac) was partitioned and two partitioned extracts, the hexane (Hex-Ac) extract and dichloromethane (Dic-Ac) extract, were obtained. Six ingredients, Ac-1 – Ac-6, were isolated by gel chromatography and were further identified their chemical structures. However, only Dic-Ac extract was tested for the toxicity and biological activities. This makes the whole structure of the experimental design confusing. I suggest that the authors modify the research structure to make this paper focus on Dic-Ac extract.
Specific comments
- Why this study only partitioned HE-Ac extract by hexane and dichloromethane? Why not the polar part?
- Because the research content for the toxicity and biological activities was only for Dic-Ac extract, I recommend the authors to modify the research structure to make this paper focus on Dic-Ac extract. Accordingly, include the title, chemical isolation and identification, etc. should be revised, and to delete the content related to Hex-Ac extract.
- If the authors want to emphasize that this plant is an important source of norcucurbitacins, as the title indicates, the comparison data for the content of norcucurbitacins of this plant with other plants should be provided in the text.
- The genus name of the plant name must use the full name when it first appears, and the abbreviation should be used when it appears later, please correct (Line 46, 48, 525).
- Line 74: The unit “70° GL” is not a general used format. I suggest to use the scientifically used unit "%".
- Line 87: The weight “126 mg” is different with the value listed in Scheme 1. Please check.
- Scheme 1 and Scheme 2: The number “8,3 g” should be “8.3 g”.
- The units of minute and hour should use the abbreviation “min” and “h”.
- Line 234: I suggest to revise the number “25 x 106” to “2.5 x 107” to follow the scientific notation rule.
- Line 419: The “Table 5” should be “Table 4”.
- Table 4: Please confirm whether the differences in the various parameters in this table are all not significant (p < 0.05).

Author Response
Dear reviewer, we appreciate the suggestions.
Response to the reviewer’s comments:
Reviewer's remarks: This work investigated the chemical composition, toxicity, and pharmacological potential, including anti-inflammatory and anti-tumor activity, of Apodanthera congestiflora. The hydroethanolic extract of its stem (HE-Ac) was partitioned and two partitioned extracts, the hexane (Hex-Ac) extract and dichloromethane (Dic-Ac) extract, were obtained. Six ingredients, Ac-1 – Ac-6, were isolated by gel chromatography and were further identified their chemical structures. However, only Dic-Ac extract was tested for the toxicity and biological activities. This makes the whole structure of the experimental design confusing. I suggest that the authors modify the research structure to make this paper focus on Dic-Ac extract
Author´s response: dear reviewer, we redefined the structural design of the article and focused on the work on the isolation of compounds and on the pharmacological activities of the dichloromethane phase. We also added, in the introduction, information regarding studies with cucurbitacins and their anti-cancer and anti-inflammatory properties.
Specific comments
- Why this study only partitioned HE-Ac extract by hexane and dichloromethane? Why not the polar part?
Author´s response: we made the correction in the text of the article. In the partition we obtain the nonpolar and polar phases. However, in this part of the research we focus on the bioguided study of the dichloromethane phase. We are working with the other phases and, later, we will publish our results.
- Because the research content for the toxicity and biological activities was only for Dic-Ac extract, I recommend the authors to modify the research structure to make this paper focus on Dic-Ac extract. Accordingly, include the title, chemical isolation and identification, etc. should be revised, and to delete the content related to Hex-Ac extract.
Author´s response: we remove the content related to the hexane phase and leave only the data obtained from the research with the dichloromethane phase. We have made all the necessary modifications so that the data is clear.
- If the authors want to emphasize that this plant is an important source of norcucurbitacins, as the title indicates, the comparison data for the content of norcucurbitacins of this plant with other plants should be provided in the text.
Author´s response: we changed the title because data on the cucurbitacin content of plants are rare. We will even be one of the first authors to report information on the content of cucurbitacins in a plant species. We use an 1H NMR tool called TopicSpin Eretic and calculate the content of the cayaponoside C5b in the dichloromethane extract.
- The genus name of the plant name must use the full name when it first appears, and the abbreviation should be used when it appears later, please correct (Line 46, 48, 525).
Author´s response: revised
- Line 74: The unit “70° GL” is not a general used format. I suggest to use the scientifically used unit "%".
Author´s response: revised
- Line 87: The weight “126 mg” is different with the value listed in Scheme 1. Please check.
Author´s response: scheme 1 has been withdrawn.
- Scheme 1 and Scheme 2: The number “8,3 g” should be “8.3 g”.
Author´s response: revised
- The units of minute and hour should use the abbreviation “min” and “h”.
Author´s response: revised
- Line 234: I suggest to revise the number “25 x 106” to “2.5 x 107” to follow the scientific notation rule.
Author´s response: revised
- Line 419: The “Table 5” should be “Table 4”.
Author´s response: revised
- Table 4: Please confirm whether the differences in the various parameters in this table are all not significant (p < 0.05).
Author´s response: a new statistical treatment was performed, this time using a t test. Consequently, there was a statistical difference in the values of glucose and cholesterol.

Round 2
Reviewer 3 Report
The article has been amended according to the comments. Although it still has some minor errors, the paper has been able to be accepted for publication in this journal. The points need to revise further are as follows:
- Line 45: the word “anti-anti-tumor activity” seems to be mistake.
- Eq. 1-3: the unit “%” should be added after “100”.
- Figure 2: The meaning of “***” should be described in its legend.
